# Irradiated Triple-Negative Breast Cancer Co-Culture Produces a Less Oncogenic Extracellular Matrix

**DOI:** 10.3390/ijms23158265

**Published:** 2022-07-27

**Authors:** Elizabeth Brett, Michael Rosemann, Omid Azimzadeh, Andrea Pagani, Cosima Prahm, Adrien Daigeler, Dominik Duscher, Jonas Kolbenschlag

**Affiliations:** 1Department of Hand, Plastic, Reconstructive and Burn Surgery, BG-Unfallklinik Tuebingen, University of Tuebingen, Schnarrenbergstraße 95, 72076 Tübingen, Germany; zab.brett@gmail.com (E.B.); cprahm@bgu-tuebingen.de (C.P.); adaigeler@bgu-tuebingen.de (A.D.); 2Helmholtz Center München, Institute of Radiation Biology, Ingolstädter Landstraße 1, Deutsches Forschungszentrum für Gesundheit und Umwelt (GmbH), 85764 Neuherberg, Germany; rosemann@helmholtz-muenchen.de (M.R.); oazimzadeh@bfs.de (O.A.); 3Department of Orthopedics, Traumatology and Hand Surgery, Hospital of Bolzano—SABES, Lorenz-Böhler-Straße 5, 39100 Bolzano, Italy; andreapagani.md@gmail.com

**Keywords:** collagen VI, irradiation, triple-negative breast cancer, invasion, tumor microenvironment

## Abstract

Triple-negative breast cancer is the most common and most deadly cancer among women. Radiation is a mainstay of treatment, administered after surgery, and used in the hope that any remaining cancer cells will be destroyed. While the cancer cell response is normally the focus of radiation therapy, little is known about the tumor microenvironment response after irradiation. It is widely reported that increased collagen expression and deposition are associated with cancer progression and poor prognosis in breast cancer patients. Aside from the classical fibrotic response, ratios of collagen isoforms have not been studied in a radiated tumor microenvironment. Here, we created one healthy co-culture of stromal fibroblasts and adipose-derived stem cells, and one triple-negative breast cancer co-culture, made of stromal fibroblasts, adipose derived stem cells, and triple-negative breast cancer cells. After irradiation, growth and decellularization of co-cultures, we reseeded the breast cancer cells for 24 h and analyzed the samples using mass spectrometry. Proteomic analysis revealed that collagen VI, a highly oncogenic collagen isoform linked to breast cancer, was decreased in the irradiated cancer co-culture. This indicates that the anti-cancer impact of radiation may be not only cell ablative, but also influential in creating a less oncogenic microenvironment.

## 1. Introduction

Breast cancer is the most common malignancy in women worldwide. Of the 2.3 million breast cancer diagnoses in 2020, 685,000 cases were fatal [1]. Given the heterogeneity of this pathology, the therapeutic approaches vary significantly between patients. 

After the St. Gallen International Breast Cancer Conference in 2013, Goldhirsch et al. highlighted a new definition of breast cancer subtypes: Luminal A and B (ER/PR^+^, HER2^−^, Ki67^+^), HER2^+^ B2 (ER/PR^+^, HER2 overexpression) HER2 overexpression (ER/PR^−^, HER2 overexpression), basal-like TNBC (ER/PR^−^, HER2^−^) and other special sub-groups [2]. Above all, triple-negative breast cancer (TNBC) has a poorer prognosis compared with the other subtypes, and mostly occurs in premenopausal young women [3]. Due to the absence of an estrogen receptor (ER), progesterone receptor (PR) and human epidermal growth factor receptor 2 (HER2), chemo-, immuno- and radiation therapy (RT) are the only effective treatments for this type of cancer, especially in combination for the treatment of stage I and II TNBCs [4]. While Collignon et al. showed that Bevacizumab combined with other drugs is a reliable option as immunotherapy in TNBC patients [5], other authors reported that RT in combination with chemotherapy provide a real benefit to patients by decreasing locoregional recurrences and increasing overall survival [4,6].

During ionizing radiation therapy, the double helix of DNA is broken, and the free radicals produced result in irreversible cell damage and death [7]. Although the molecular effects of radiotherapy on tumor cells in the treatment of early stage breast cancer have been amply demonstrated, little is known about the response of the tumor microenvironment (TME) to irradiation. The invasive front of TNBC tumors is built with linear extensions of collagen which form road-like structures radiating perpendicularly from the tumor border. This particular composition of the border promotes tumor expansion and intravasation [8]. Our group has previously interrogated the formation of an oncogenic extracellular matrix (ECM) using an in vitro model, and observed a linearized matrix comprised chiefly of oncogenic collagen type VI [8]. The purpose of this study is to disrupt the model using ionizing radiation so to understand the ECM produced by a co-culture which otherwise creates an oncogenic ECM. Increased collagen expression and deposition is associated with cancer progression and poor prognosis. Aside from the classical fibrotic reaction, the production of collagen by the TME has not been studied in a radiated cancer microenvironment.

In this study, we highlight that the anti-cancer impact of radiation may be not only cell ablative, but also influential in creating a microenvironment that is inherently less oncogenic. Here, we deepen the current knowledge regarding collagen VI within the tumor microenvironment in order to guide further breast cancer research, diagnosis and treatment.

## 2. Results

### 2.1. One Dose of 5 Gy Does Not Alter the Viability of the Cells

Irradiation of the co-culture groups A and B, along with ASCs alone, fibroblasts alone and MDA-MB-231 alone with one dose of 5 Gy did not significantly affect cell viability (Figure 1), allowing for further downstream testing with the same dosage and culture pattern. Radiation therapy had no significant impact on survival rates, and the proliferative capacity was comparable between the five different groups.

Proportional bar graphs in the figure below (Figure 1) highlight that TNBC co-culture (Group A) and fibroblasts alone have a cell viability of approximately 97% in irradiated groups. This value is slightly increased when compared with the non-irradiated groups, although not significantly. This reduced viability trend of irradiated cells could decrease further by increasing the irradiation power or the number of radiant doses. The next stage was to test the production of collagen VI.

### 2.2. Irradiated Co-Culture (Fibroblasts, ASCs, and MDA-MB-231) Produces Matrix with Decreased Collagen Vi Isoforms

In order to evaluate the production of collagen, the co-culture groups were seeded, irradiated with a single shot of 5 Gy and cultured for 7 days (Figure 3). One week later, the matrix was decellularized, then recellularized with MDA-MB-231. After one day of culture, the cell/matrix combination was prepared for mass spectrometry. Proteomic analyses showed that levels of Collagen VIA3 were significantly higher in the non-irradiated TNBC matrix compared to the non-irradiated healthy matrix.

Although the three strands of the collagen VI isoforms (COL6A1, COL6A2 and COL6A3) were more abundant in the cancer matrix compared to the healthy matrix in the irradiated groups, these differences were not significant in our Volcano Plots (Figure 2). Radiation therapy therefore blocked the production of collagen VI in TNBC co-cultures.

## 3. Discussion

Solid tumors are composed of malignant cells surrounded by an “echological niche” that stimulates cancer progression and resistance [9,10]. The cellular component of the TME is represented by cancer-associated fibroblasts (CAFs), mesenchymal stromal cells (MSCs), endothelial cells (ECs), pericytes and immune cells [11]. Krisnawan et al. highlighted that Radiotherapy can impact and transform the cellular component of the TME by inducing a rapid loss of hyaluronic acid within the ecological niche and altering CAFs function, which is a central protagonist of collagen production [10].

Compared to other types of breast cancer, TNBC has limited treatment options, is prone to recurrence and metastasis and has a poor prognosis. Studies have documented that the incidence of locoregional recurrence in TNBC patients peaks during years 1–4, and that individuals are more likely to have locoregional recurrence in comparison with distant metastasis [12,13]. Irradiation frequently accompanies chemotherapy drugs such as taxane, anthracycline, cyclophosphamide, cisplatin and fluorouracil in the treatment of TNBC, and when performed, is associated with increased survival rates of patients [14]. To focus solely on the cellular response to radiation is to ignore a massive feature of TNBC: the tumor matrix collagen content and physicality.

By affecting the adjacent tumor microenvironment, radiotherapy has also a key impact on tumor vascularization. RT destroys tumor vascularization, increasing the area of tumor hypoxia [15,16]. By inducing an inflammatory reaction, an hypoxic context reduces oxygen-dependent DNA damage and weakens the anti-cancer RT effect [17]. Hypoxia induces HIF–1 alpha-mediated cell survival, increasing glycolysis, the accumulation of pyruvate and lactate and increasing tumor resistance [18]. Altogether, hypoxia promotes tumor angiogenesis and tolerance. Cancer cells are therefore much more resistant to RT in hypoxic conditions.

It seems that a low dose of irradiation leads to cell death by promoting apoptosis but is definitely unable to induce an effective anti-tumor response. On the other hand, a high dose can destroy the healthy cellular component of the cancer, generating a dangerous in situ “vaccination effect” which modulates the adaptive response of the immune cells of the TME [19] and, in the end, induces radioresistance. The protocols for delivering small daily radiotherapy doses can exploit tumor cells’ vulnerability in repairing DNA damage, and spare healthy cells, giving them the possibility to activate their repair mechanisms [20]. This increases the therapeutic ratio compared to single-shot delivery. The majority of solid tumors need a 50 to 70 Gy total radiation dose, most commonly fractionated in 1.8 to 2 Gy doses. Doses of radiation for TNBC range from 1000–5000 cGy [21], and are fractionated over the course of five to six weeks. The cells in the irradiation data of this research received a one-time dose of 5 Gy, consistent with the literature [22], which did not significantly affect the viability of the cells. Considering the anti-cancer effect of irradiation, it is interesting to observe that the irradiated cells themselves tend to produce a matrix which is lower in collagen VI than its non-irradiated comparison. This highlights the influence of radiation on the cells in making a matrix which is less oncogenic than its non-irradiated comparison.

ASCs alone have been well described as an adjunct cell-therapy option in radiation oncology and regenerative medicine. Baaße et al. [23] highlighted that ASCs isolated from breast tissue are robust in their response to irradiation, as a result of early cell cycle arrest and rapid DNA damage repair. This characteristic explains the high therapeutic efficacy of fresh ASCs in grafted fat under irradiated skin, using paracrine influence to rescue the fibrotic dermis caused by chronic radiation damage [24]. Considering the restorative impact of stem cells within an irradiated TME, and the data within this study, the use of only triple-negative breast cancer cells and a two-dimensional culture model is an important limitation of our study. Bar graphs show that irradiated TNBC co-culture (Group A) and fibroblasts alone have a slightly increased cell viability of 97% when compared with non-irradiated groups, although not significantly. Using a different irradiation protocol, changing radiation doses or increasing the number of radiation treatments (e.g., at day 1, 3 and 5 after seeding) could make this difference significant. All these elements represent future research focuses.

We previously confirmed that collagen VI is produced by fibroblasts in response to a paracrine co-culture of adipose-derived stem cells and MDA-MB-231, which, together, secrete high levels of the chemokine CCL5 [8]. A valuable next experiment would be to test the CCL5 production of an irradiated co-culture of ASCs/MDA-MB-231, to investigate if this stage in the linear matrix deposition is impacted by irradiation. After all, a reduction in CCL5 means a decrease in a sequela of pro-oncogenic events, which is known to be a result of radiation in TNBC [25].

Altogether, this study illustrated that irradiated cells in the TNBC microenvironment produce ECM which contains lower proportions of oncogenic collagen VI. This finding, if confirmed in an in vivo model, indicates the ancillary benefit of radiation therapy in creating a remodeled ECM which is inherently less oncogenic. This implication has further value when applied to radiotherapy of all cancers, beyond mammary carcinoma.

## 4. Methods and Materials

### 4.1. Cell Lines and Cell Cultures

TNBC cells (MDA-MB-231), human fibroblasts (HS-27) from Lonza and Adipose-Derived Stem Cells (ASCs) (Poietics™, donors 29635, 31363) were all purchased as cell lines from Lonza and used in the downstream experiments. Cells stocks were created in the early stages of culture to maintain passage <15. Normal culture conditions included a humidified chamber at 37 °C with 5% CO_2_. All cell culture work in this project was carried out at these parameters. MDA-MB-231 and fibroblasts (HS-27) were cultured in DMEM (Dulbecco’s modified Eagle’s medium), 10% fetal bovine serum and 1% PenStrep (henceforth referred to as “full media”). ASCs were specifically cultured in StemMACS rather than DMEM, with 10% FBS and 1% PenStrep. Group A (triple-negative breast cancer co-culture) presented all the cell types in equal number (9.6 cm^2^ = 50,000 total cells). Group B (healthy co-culture) had 25,000 HS-27 and 25,000 ASCs per 9.6 cm^2^.

### 4.2. Cell Viability—Live/Dead Staining

In order to analyze cell viability in non-irradiated and irradiated cultures, cells were separated per cell co-culture and cell monocultures. Cells were seeded in five different cultures and irradiated after 24 h with one dose of 5 Gy. Living/dead cells were detected with Calcein AM (1 μg/mL) and propidium iodide (2 μg/mL). The Live/dead staining Kit (Thermo Fisher Scientific, Waltham, MA, USA) was used according to the manufacturer’s instructions. Photographs were taken with an Axio Observer Z.1 fluorescence microscope (Carl Zeiss AG, Oberkochen, Germany). Proportional bar graphs were performed with Microsoft Excel version 16.0 (Microsoft Corporation, Redmond, WA, USA).

### 4.3. Irradiation of Cells and Decellularization

In order to assess ECM deposition by irradiated cells, cells were seeded in six-well plates, as described in Figure 3, and cultured for one day to ensure attachment. Plates were then irradiated with one dose of 5 Gy in a warmed chamber, and afterwards cultured per the normal protocol of 7 days with addition of 50 mm of ascorbic acid. Viability assays were run 24 h after irradiation, and decellularization of ECM followed thereafter. Specifically, co-cultures were washed with phosphate-buffered saline (PBS), and decellularized using a 20 mm NH_4_OH/1% Triton-X solution. The decellularized solution was then diluted with PBS containing 1% PenStrep and stored at 4 °C overnight.

### 4.4. Cell/Matrix Preparation for Mass Spectrometry and Proteomic Analysis

ECMs derived from both groups A and B were prepared over the week-long experimental phase (Figure 3), reseeded with 10,000 MDA-MB-231 cells and cultured for 24 h. Afterwards, the cell/matrix cultures were homogenized in RIPA buffer and protease inhibitor. Samples were kept on ice until filter-aided sample preparation (FASP) on the same day. FASP digest was run on samples of 10 µg cell/matrix extracts per the existing standard protocol [26]. Progenesis QI for proteomics analysis provided quantitative protein readout [27]. Molecular signaling was interpreted via gene ontology term analysis (molecular function) with Ingenuity Pathway Analysis (IPA, Qiagen, 19300 Germantown MD 20874, USA) [28].

### 4.5. Statistical Analysis

Statistical analyses were performed with GraphPad Prism (GraphPad Software, Inc., San Diego, CA, USA). Two tailed *t*-test were used to show any significant differences of Collagen VI in either irradiated/non-irradiated cell populations. A *p*-value of <0.05 was considered significant.

## Figures and Tables

**Figure 1 ijms-23-08265-f001:**
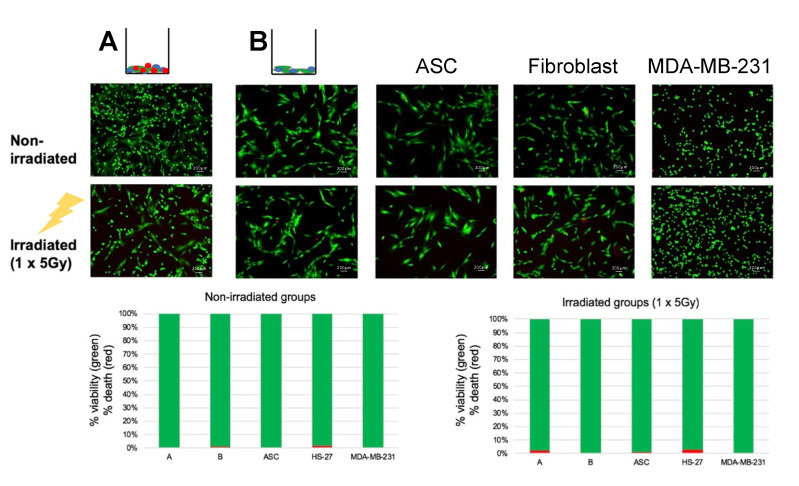
**Analysis of radiation on cell viability.** Top: Fluorescent micrographs of cells (group A and B) in non-irradiated and irradiated groups, separated per cell co-culture and cell monocultures. Below, proportional bar graphs showing quantified live/dead cells from irradiation. TNBC co-culture (Group A) and fibroblasts alone showed cell viability of approximately 97% in irradiated groups.

**Figure 2 ijms-23-08265-f002:**
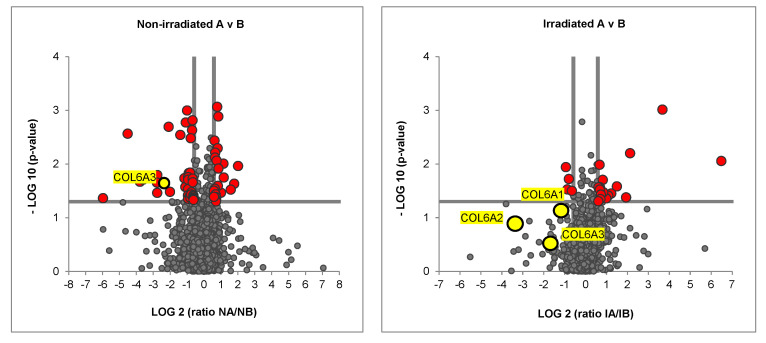
**Proteomic Analysis of non–irradiated and irradiated co-cultures.** Volcano plot showing significantly differently abundant proteins in irradiated TNBC matrix versus irradiated healthy matrix (red data points). Three yellow data points below the horizontal line of significance represent the three strands of collagen VI, COL6A1, COL6A2 and COL6A3.

**Figure 3 ijms-23-08265-f003:**
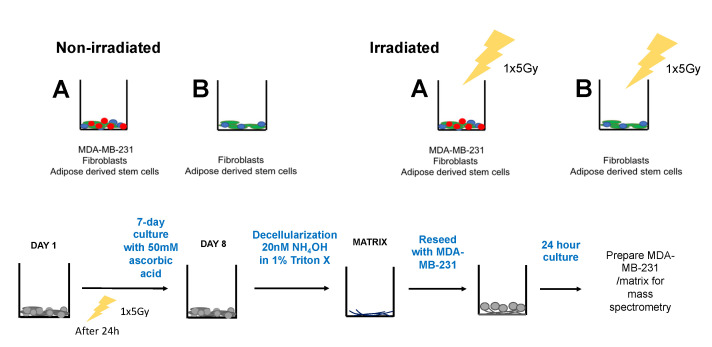
**Schematic showing workflow for irradiated cell groups.** Cells (group A and B) were seeded at DAY 1 and irradiated 24 h after with one dose of 5 Gy. Cells were cultured for one week, decellularized and recellularized with MDA-MB-231. One day later, samples were prepared for mass spectroscopy. Group A: MDA-MB-231, Fibroblasts, ASCs; B: Fibroblasts, ASCs.

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
