# Peer review of "Irradiated Triple-Negative Breast Cancer Co-Culture Produces a Less Oncogenic Extracellular Matrix"

_ijms, 2022, doi:10.3390/ijms23158265_

Round 1
Reviewer 1 Report
The study is an in vitro study using triple-negative breast cancer cells, fibroblast, and adipose-derived stem cells. Authors have designed the study to analyze proteomic profiles in radiated and nonradiated cancer cells. They have created one healthy co-culture of stromal fibroblasts and adipose-derived stem cells, and one breast cancer co-culture, made of stromal fibroblasts, adipose-derived stem cells, and triple-negative breast cancer cells.The authors irradiated cells and after 24 hours, reseeded the breast cancer cells for 24 hours and analyzed them by mass spectrometry. Proteomic analysis revealed a decrease in collagen VI levels, an important oncogenic type of collagen. The authors concluded that irradiation may provide a potential antitumoral benefit also regarding tumor microenvironment creating a less oncogenic microenvironment.
The study fits a communication type of article as only cell culture and proteomics were performed.
I recommend changing the title of the article, including the name of the cancercells used. Suggest include gene expression analysis for 6A1.
There are some minor english errors that should be reviwed.
Author Response
Response to Reviewer 1 Comments
Review report 1:
The study is an in vitro study using triple-negative breast cancer cells, fibroblast, and adipose-derived stem cells. Authors have designed the study to analyze proteomic profiles in radiated and nonradiated cancer cells. They have created one healthy co-culture of stromal fibroblasts and adipose-derived stem cells, and one breast cancer co-culture, made of stromal fibroblasts, adipose-derived stem cells, and triple-negative breast cancer cells. The authors irradiated cells and after 24 hours, reseeded the breast cancer cells for 24 hours and analyzed them by mass spectrometry. Proteomic analysis revealed a decrease in collagen VI levels, an important oncogenic type of collagen. The authors concluded that irradiation may provide a potential antitumoral benefit also regarding tumor microenvironment creating a less oncogenic microenvironment. The study fits a communication type of article as only cell culture and proteomics were performed.
I recommend changing the title of the article, including the name of the cancer cells used. Suggest including gene expression analysis for 6A1. There are some minor English errors that should be reviewed.
Response 1:
Thanks for reviewing our article. We are happy that you have appreciated our work. Since we also believe that the title and name of the tumor cell cultures should be modified, we changed the title and the nickname of the cancer cell co-cultures. The name "TNBC co-culture" has taken the place of the generic name "breast cancer co-culture" to emphasize the cell line, while maintaining a short and suggestive name. Collagen VI gene expression has always been an important focus in our research in this area. A primordial gene expression analysis was already performed during our in-vitro studies. However, the final results will be published in a separate article in the future. Finally, all the text has been revised from the linguistic point of view.
Reviewer 2 Report
1. The issue of the influence of the cellular microenvironment on the radioresistance and phenotype of tumor cells is an important challenge in clinicals and radiobiology. The presented manuscript discusses the anti-cancer impact of radiation on cellular microenvironment. The manuscript is primarily based on one co-author Dr. Elizabeth Anne Brett' Ph.D. thesis "Evidence of an unreported cell relationship responsible for linear collagen of triple negative breast tumors" (2020) (https://mediatum.ub.tum.de/doc/1547730/file.pdf)
2. I do believe that it is essential that the main results have already been published in the paper: Brett, E., Sauter, M., Timmins, É., Azimzadeh, O., Rosemann, M., Merl-Pham, J., ... & Duscher, D. (2020). Oncogenic linear collagen VI of invasive breast cancer is induced by CCL5. Journal of clinical medicine, 9(4), 991. https://www.mdpi.com/2077-0383/9/4/991 3.1. Despite the type of the manuscript (Communication), the methods used are insufficient to justify the conclusions. 3.2. Can the сhanges in collagen synthesis be explained by a decrease of the metabolic activity of the irradiated cells? I suppose it's up for discussion.3.3. I don't understand how the authors used statistical analysis. Which criteria were used? In the PhD Thesis, it's a lot clearer.
3.4. The references list contains primarily 2001-2015 years sources. The authors should revise the references list with the next related papers:
- Krisnawan, V. E., Stanley, J. A., Schwarz, J. K., & DeNardo, D. G. (2020). Tumor microenvironment as a regulator of radiation therapy: new insights into stromal-mediated radioresistance. Cancers, 12(10), 2916.
- Jarosz-Biej, M., Smolarczyk, R., Cichoń, T., & Kułach, N. (2019). Tumor microenvironment as a “game changer” in cancer radiotherapy. International journal of molecular sciences, 20(13), 3212.
- Girigoswami, K., Saini, D., & Girigoswami, A. (2021). Extracellular matrix remodeling and development of cancer. Stem Cell Reviews and Reports, 17(3), 739-747.
- Tuieng, R. J., Cartmell, S. H., Kirwan, C. C., & Sherratt, M. J. (2021). The Effects of Ionising and Non-Ionising Electromagnetic Radiation on Extracellular Matrix Proteins. Cells, 10(11), 3041.
Author Response
Irradiated TNBC co-culture produces a less oncogenic extracellular matrix
Elizabeth Brett1, Michael Rosemann2, Omid Azimzadeh2, Andrea Pagani3, Cosima Prahm1, Adrien Daigeler1, Dominik Duscher1*, Jonas Kolbenschlag1*
Response to Reviewer 1 Comments
Point 1: The issue of the influence of the cellular microenvironment on the radioresistance and phenotype of tumor cells is an important challenge in clinicals and radiobiology. The presented manuscript discusses the anti-cancer impact of radiation on cellular microenvironment. The manuscript is primarily based on one co-author Dr. Elizabeth Anne Brett' Ph.D. thesis "Evidence of an unreported cell relationship responsible for linear collagen of triple negative breast tumors" (2020) (https://mediatum.ub.tum.de/doc/1547730/file.pdf).
Response 1:
Thank you very much for reviewing our manuscript. As you mentioned, the Ph. D. thesis of Dr. Brett “Evidence of an unreported cell relationship responsible for linear collagen of triple negative breast tumors” is the reference point for the submitted communication. In this paper we moved forward highlighting the response of the tumor microenvironment (TME) to radiation therapy. How the cells answer and which molecules they are producing after radiation is the focus of this communication, hence we investigated the role of TME from a cellular point of view by using mass spectrometry.
Point 2: I do believe that it is essential that the main results have already been published in the paper: Brett, E., Sauter, M., Timmins, É., Azimzadeh, O., Rosemann, M., Merl-Pham, J., ... & Duscher, D. (2020). Oncogenic linear collagen VI of invasive breast cancer is induced by CCL5. Journal of clinical medicine, 9(4), 991. https://www.mdpi.com/2077-0383/9/4/991.
Response 2:
The article “Oncogenic linear collagen VI of invasive breast cancer is induced by CCL5” can be considered as the foundation for this paper. Here we started exploring the response of the tumor microenvironment to radiation therapy, paying particular attention to the production of collagen VI.
Point 3.1: Despite the type of the manuscript (Communication), the methods used are insufficient to justify the conclusions.
Response 3.1:
We thank the reviewer for their comment. The reviewer has correctly noted that this is a communication, and not a full research manuscript. We conclude from this preliminary finding that the lower content of collagen type VI resulting from radiation is likely one of the reasons why radiation limits cancer development. It would be helpful for the reviewer to pinpoint exactly which conclusion(s) they find unsupported.
Point 3.2: Can the сhanges in collagen synthesis be explained by a decrease of the metabolic activity of the irradiated cells? I suppose it's up for discussion.
Response 3.2:
We thank the reviewer for this comment. Radiation has a global effect on targeted cells which impacts viability, mitosis, DNA methylation, biosynthesis... it is indeed an evolving discussion and one to which we are hoping this communication can add some knowledge.
Point 3.3: I don't understand how the authors used statistical analysis. Which criteria were used? In the PhD Thesis, it's a lot clearer.
Response 3.3:
We thank the reviewer for this very thorough analysis of our body of work. We have altered the Statistical Analysis section of Materials and Methods to include: 'Two tailed T-tests were used to show any significant differences of collagen VI in either irradiated/non-irradiated cell populations.
Point 3.4: The references list contains primarily 2001-2015 years sources. The authors should revise the references list with the next related papers:
- Jarosz-Biej, M., Smolarczyk, R., Cichoń, T., & Kułach, N. (2019). Tumor microenvironment as a “game changer” in cancer radiotherapy. International journal of molecular sciences, 20(13), 3212.
- Girigoswami, K., Saini, D., & Girigoswami, A. (2021). Extracellular matrix remodeling and development of cancer. Stem Cell Reviews and Reports, 17(3), 739-747.
- Tuieng, R. J., Cartmell, S. H., Kirwan, C. C., & Sherratt, M. J. (2021). The Effects of Ionising and Non-Ionising Electromagnetic Radiation on Extracellular Matrix Proteins. Cells, 10(11), 3041.
Response 3.4:
Thank you for making this comment because it allowed us to improve the "discussion" section of our manuscript. In accordance with the recommended articles, we analysed the suggested papers and expanded the discussion by citing recent works by Jarosz-Biej et al, Krisanawan et al, Klein et al and Tsoutsou et al.
“Solid tumors are composed of malignant cells surrounded by an “echological niche” that stimulate cancer progression and resistance (12, 13). The cellular component of the TME is represented by Cancer associated fibroblasts (CAFs), Mesenchymal stromal cells (MSCs), Endothelial cells (ECs), Pericytes and Immune cells (14). Krisnawan et al. highlited that Radiotherapy can impact and transform the cellular component of the TME by inducing a rapid loss of hyaluronic acid within the echological niche and altering CAFs function, which is a central protagonist of collagen production (13).”
“By affecting the adjacent Tumor microenvironment, Radiotherapy has also a key impact on tumor vascularization. RT destroys tumor vascularization increasing the area of tumor hypoxia (18, 19). By inducing an inflammatory reaction, an hypoxic context reduces oxygen-dependent DNA damage and weakens the anti-cancer RT effect (20). Hypoxia induces HIF-1 alpha mediated cell survival, increasing glycolysis, the accumulation of pyruvate and lactate and increasing tumor resistance (21). Alltogether, hypoxia promotes tumor angiogenesis and tolerance. Cancer cells are therefore much more resistant to RT in hypoxic conditions.”
“It seems that a low dose of Irradiation leads to cell death by promoting apoptosis but is definitely unable to induce an effective antitumor response. On the other side, a high dose can destroy the healthy cellular component of the cancer generating a dangerous in situ “vaccination effect” which modulates the adaptive response of the immune cells of the TME (22) and induce, in the end, radioresistance. The protocols for delivering small daily Radiotherapy doses can exploit tumor cells’ vulnerability in repairing DNA damage, and spare healthy cells, giving them the possibility to activate their repair mechanisms (23). This to increase the therapeutic ratio compared to single-shot delivery. The majority of solid tumors need 50 to 70 Gy total radiation dose, most commonly fractionated in 1.8 to 2 Gy doses.”
Round 2
Reviewer 2 Report
The authors have taken into account all my previously comments and corrected the manuscript.
Today I found that the authors were referring to the 2020 healthcare statistics (line 33). I recommend updating the statistics to 2021.
I guess the manuscript may be published in the present form after minor corrections.